# Utilizing Osteocyte Derived Factors to Enhance Cell Viability and Osteogenic Matrix Deposition within IPN Hydrogels

**DOI:** 10.3390/ma13071690

**Published:** 2020-04-04

**Authors:** Laurens Parmentier, Mathieu Riffault, David A. Hoey

**Affiliations:** 1Trinity Centre for Biomedical Engineering, Trinity Biomedical Sciences Institute, Trinity College Dublin, Dublin 2 D02 R590, Ireland; 2Department of Mechanical and Manufacturing Engineering, School of Engineering, Trinity College Dublin, Dublin 2 D02 DK07, Ireland; 3Polymer Chemistry and Biomaterials Group, Centre of Macromolecular Chemistry (CMaC), Department of Organic and Macromolecular Chemistry, Ghent University, 9000 Ghent, Belgium; 4Advanced Materials and Bioengineering Research Centre, Trinity College Dublin & RCSI, Dublin 2 D02 VN51, Ireland

**Keywords:** bone, collagen type I, alginate, conditioned medium, viability, MSC, osteogenesis

## Abstract

Many bone defects arising due to traumatic injury, disease, or surgery are unable to regenerate, requiring intervention. More than four million graft procedures are performed each year to treat these defects making bone the second most commonly transplanted tissue worldwide. However, these types of graft suffer from a limited supply, a second surgical site, donor site morbidity, and pain. Due to the unmet clinical need for new materials to promote skeletal repair, this study aimed to produce novel biomimetic materials to enhance stem/stromal cell osteogenesis and bone repair by recapitulating aspects of the biophysical and biochemical cues found within the bone microenvironment. Utilizing a collagen type I–alginate interpenetrating polymer network we fabricated a material which mirrors the mechanical and structural properties of unmineralized bone, consisting of a porous fibrous matrix with a young’s modulus of 64 kPa, both of which have been shown to enhance mesenchymal stromal/stem cell (MSC) osteogenesis. Moreover, by combining this material with biochemical paracrine factors released by statically cultured and mechanically stimulated osteocytes, we further mirrored the biochemical environment of the bone niche, enhancing stromal/stem cell viability, differentiation, and matrix deposition. Therefore, this biomimetic material represents a novel approach to promote skeletal repair.

## 1. Introduction

Bone defects arise due to traumatic injury, disease, or orthopedic surgeries and many of these are unable to regenerate requiring intervention [1]. More than four million graft procedures are performed each year to treat these defects making bone the second most commonly transplanted tissue worldwide [2]. The prevalence of bone defects, their associated morbidity, and socio-economic cost necessitates the need for new materials to promote skeletal repair through regeneration [3]. It is intuitive that a material which mimics the native tissue may be optimal to promote repair and it is autografts that are currently used as the clinical gold standard. However, these types of grafts suffer from a limited supply, a second surgical site, donor site morbidity, and pain [4]. Allografts on the other hand carry the risk of transmission of infections or an enhanced immune response [5,6]. Therefore, bone substitute materials are gaining much interest as an alternative to grafts. These materials should be designed in such a way so that they provide both biochemical and biophysical cues similar to that found in the native extracellular matrix (ECM) inductive of bone and the marrow cavity, that can promote cell attachment, migration, proliferation, and osteogenesis [7].

There are numerous biochemical and biophysical cues present within the bone niche which promote progenitor cell osteogenesis. For example, physical cues derived from the substrate can dramatically influence cell behavior. Substrate stiffness in the range of 25–40 kPa has been shown to be osteogenic mimicking the non-mineralized osteoid tissue deposited by osteoblasts during bone formation [8]. The surface topography presented to a cell is also a mediator for stem cell fate since it influences the extent a cell can spread over the surface. A higher degree of spreading commits the mesenchymal stromal/stem cell (MSC) to the osteogenic lineage [9,10,11,12]. Furthermore, a more disordered surface topography results in a higher osteogenic stimulus [13]. In addition, an average roughness on the micrometer range has been found to be the most favorable for the osteogenic differentiation of MSCs [14]. The biochemical makeup of the material can also greatly influence cell behavior. The native composition of bone tissue is mainly comprised of collagen type I, interspersed with mineral crystals [15]. Collagen type I, the major protein present in the tissue, comprises a number of adhesive ligands which influences the degree of cell attachment, spreading and hence the osteogenic differentiation of resident cells [9,10]. Therefore, there is a need for a synthetic alternative to autograft which aspires to recapitulate the various structural and physical properties of bone ECM in a 3D environment [16].

In addition to biophysical cues derived from the cell substrate, biochemical cues within the bone niche arise predominately from resident cells via paracrine signaling. Bone is exquisitely mechanosensitive and can structurally adapt to its mechanical environment to maintain function [17]. Macroscopic loading resulting in tissue deformation leads to fluid flow at the cellular level in the lacuno-canalicular system [18]. The fluid flow translates in various stimuli at the cell level but it has been shown that stimulation through a fluid-flow induced shear stress has the greatest capacity for mechanical activation and prevention of apoptosis [19,20,21]. Osteocytes are the most abundant cell type within bone and are largely believed to be the master orchestrator of bone physiology [22]. Factors secreted by mature bone cells, such as osteocytes, have been shown to drive mesenchymal stem cell recruitment, proliferation, and osteogenic differentiation, with medium collected from mechanically stimulated osteocytes demonstrating a greater capacity for osteogenesis [23,24,25,26]. This therefore demonstrates that factors secreted by resident bone cells may represent novel anabolic agents to functionalize materials to enhance bone repair.

Taken together, this study aims to produce novel biomimetic materials to enhance stem/stromal cell osteogenesis and bone repair by recapitulating aspects of the biophysical (ECM structure/stiffness) and biochemical cues (osteocyte secretome) found within the bone microenvironment.

## 2. Materials and Methods

### 2.1. Interpenetrating Polymer Network (IPN) Synthesis

Interpenetrating polymer network hydrogels were made with a final rat tail collagen type I (COL-I) (Corning, Bedford, MA, USA) concentration of 1.5 mg/mL mixed in a 1:1 ratio with either a final concentration of 5 (interpenetrating polymer network (IPN) 5), 10 (IPN 10), or 15 (IPN 15) mg/mL of ultrapure, low-viscosity, and high glucuronic acid (LVG) alginate (ALG) (Novamatrix, Sandvika, Norway) in phosphate buffered saline (PBS) (Thermo Fisher Scientific, Carlsbad, CA, USA). According to the manufacturer’s instructions, collagen was brought to alkalinity to assist in the gelling procedure. The alginate stock solution was first centrifuged at 650 g for 5 min, diluted to 10, 20, and 30 mg/mL, mixed with the collagen solution and added to custom-made molds. These molds were then placed in petri dishes for collagen gelation at 37 °C for 30 min. In the meantime, 20 mM calcium chloride (Sigma, Arklow, Ireland) in ultrapure water was prepared and brought up to a pH of 7.2 using 1 N NaOH. The calcium chloride was used to ionically crosslink the alginate present in the IPN hydrogels for an additional 3 h at 37 °C. Control sample hydrogels were made that completely consisted of 1.5 mg/mL collagen type I by following the same gelation procedure as outlined above. The samples for compressive testing were stored in ultrapure water whereas the samples for histology and scanning electron microscopy (SEM) were washed with PBS, fixed with 4% paraformaldehyde (PFA) for 4 h, subjected to 30 min of 30% ethanol (EtOH) and stored in 50% EtOH.

### 2.2. Compressive Testing

Unconfined compressive testing was performed on hydrogels submersed in PBS up to 30% strain at a strain rate of 1 mm/min (Zwick Roell Z005, Herefordshire, UK, 5 N load cell). A 0.005 N pre-load was applied so that the impermeable compressive plates touch the sample on both sides. The compressive modulus was then calculated as the slope in the linear region of the stress-strain curve between 0% and 30% strain.

### 2.3. Scanning Electron Microscopy

The fixed IPN samples that were stored in 50% EtOH were further dehydrated in concentrated EtOH baths (70% EtOH, 90% EtOH, and absolute ethanol) before being transferred to a critical point dryer (Quorum Technologies, Lewes, UK) for further dehydration. The samples were then adhered to sample stubs using carbon tape, sputter coated with 5 mm of platinum-palladium (Cressington sputter coater, Cressington Scientific Instruments Ltd., Watford, UK) and imaged with a scanning electron microscope (Zeiss Ultra Plus, Cambourne, UK).

### 2.4. Cell Culture

The MLO-Y4 cell line (Kerafast, Boston, MA, USA) [27] is a murine derived osteocyte cell line that was cultured, as previously described [28], on 0.15 mg/mL rat tail collagen type I (Sigma)-coated culture flasks with α-modification of Eagle’s minimal essential medium (α-MEM) (Labtech, Heathfield, UK) supplemented with 5% fetal bovine serum (FBS) (Labtech), 5% calf serum (CS) (Sigma), 1% penicillin/streptomycin (Pen/Strep) (Sigma), and 1% L-glutamine (Sigma). Human mesenchymal stromal/stem cells (hMSCs) (Lonza, Huddersfield, UK) were cultured with low glucose Dulbecco’s modified Eagle’s medium (DMEM) (Sigma) supplemented with 10% FBS and 1% Pen/Strep. Standard incubator culture conditions (37 °C, 5% CO_2_) were used for all cell types. MLO-Y4 cells were used between passage numbers 33 and 36 and HMSCs between passage numbers 3 and 4.

### 2.5. Conditioned Medium Production

MLO-Y4 cells were seeded on 0.15 mg/mL rat tail collagen type I-coated 6-well plates at a density of 11,600 cells/cm^2^ in 2 mL of α-MEM supplemented with 2.5% FBS, 2.5% CS, 1% Pen/Strep, and 1% L-glutamine per well. After 48 h of culture, these cells were either subjected to mechanical stimulation or left in culture for an extra 2 h. For mechanical stimulation, 6-well plates were placed on an orbital shaker (Carl Roth, Karlsruhe, Germany) for 2 h at a rotational speed of 100 rpm in an incubator. Both the statically cultured as well as the mechanically activated osteocytes were then washed with 2 mL of PBS and 833 μL of serum free media was applied onto the cells. After 24 h of culture, both the static (S-CM) as well as the mechanically activated (MA-CM) conditioned medium was collected, centrifuged at 3000 g for 10 min at 4 °C to remove debris after which the supernatant was used or stored at −20 °C.

### 2.6. Conditioned Medium Osteogenesis Study

hMSCs were seeded at a density of 5000 cells/cm^2^ in 6-well plates in either 2 mL of growth medium (α-MEM + 1% Pen/Strep + 1% L-glutamine) with the addition of 10% FBS as a negative control (GM), static osteocyte conditioned medium supplemented with 10% FBS and osteogenic factors (100 nM dexamethasone (Sigma), 10 mM β-glycerol phosphate (Sigma) and 0.28 mM ascorbic acid (Sigma)) (S-CM^ost^), mechanically activated osteocyte conditioned medium with the addition of 10% FBS and osteogenic factors (MA-CM^ost^) or growth medium supplemented with 10% FBS and osteogenic factors as a positive control (GM^ost^). These four different media groups were cultured for 21 days with a medium change twice a week.

#### 2.6.1. DNA Analysis

After 21 days of culture, the hMSC monolayer was washed with cold PBS where after 100 μL of lysis buffer (DNAse free water (Thermo Fisher Scientific) with 0.2% Triton X100 (Sigma), 10 mM Tris pH8 (Thermo Fisher Scientific), and 0.5% phenylmethylsulfonyl fluoride (PMSF) (Sigma)) was added per well. Cells were then detached with a cell scraper, transferred to a microtube, vortexed, sonicated, frozen at −80 °C, thawed and homogenized with a needle. A quant-it picogreen dsDNA assay kit (Invitrogen, Carlsbad, CA, USA) was used to quantify the amount of DNA present in the cell lysate samples with reference to set standards. The samples and standards were excited at 480 nm and their emission was captured at 520 nm.

#### 2.6.2. Collagen Production

Collagen was stained and quantified through a protocol previously described [29,30]. The hMSC monolayer was fixed with 10% formalin (Sigma) for 15 min with PBS washing steps both before and after the fixation procedure. After 1 mL of picrosirius red was added per well, gentle shaking was applied for 1 h at room temperature. The cell monolayer was then washed twice with 0.5% acetic acid (Sigma) and once with ultrapure water. To quantify the amount of collagen deposited by the HMSCs, 1 mL of PBS was added per well and a cell scraper was used to remove the monolayer of cells and subsequently transferred to a microtube. After centrifugation at 14,000 g for 10 min at room temperature, the pellet was resuspended in 0.5 M NaOH and vortexed until the stain was dissolved. The samples were then compared against a collagen standard (Corning) that was centrifuged and resuspended in 0.5 M NaOH. The absorbance of both standards and samples was read at 550 nm.

#### 2.6.3. Mineral Production

Mineral deposits were stained and quantified through a protocol previously described [29,30]. The hMSC monolayer was fixed with 10% formalin for 15 min with PBS washing steps both before and after the fixation procedure. The fixed monolayers were incubated with 2 mL of alizarin red S (ARS) solution (Sigma) at a pH of 4.1 for 20 min at room temperature with gentle shaking. Afterwards, the cells were washed with excess ultrapure water until the background staining was maximally cleared. To quantify the amount of mineral that was produced by the differentiating hMSCs, 1.6 mL of 10% acetic acid was added to each well. The plates were then incubated for 30 min at room temperature while shaking at 150 rpm on an orbital shaker. The monolayer of cells were detached using a cell scraper, transferred to a microtube, vortexed, and heated to 85 °C for 10 min after which these tubes were transferred to ice for 5 min. Once the tubes were fully cooled, they were centrifuged at 20,000 g for 15 min. After adding 200 μL of 10% ammonium hydroxide to 500 μL of the supernatant, the samples were quantified against serial dilutions of ARS solution by reading their absorbance at 405 nm.

### 2.7. IPN + Conditioned Medium Osteogenesis Study

Both static as well as mechanically activated osteocyte conditioned medium were concentrated up to 10 mg/mL of protein content using a Speedvac concentrator (Savant, Carlsbad, CA, USA). The S-CM was then diluted to 0.5 mg/mL (S-CM_0.5_) whereas the MA-CM was both diluted to 5 (MA-CM_5_) and 0.5 (MA-CM_0.5_) mg/mL. All the conditioned media was then sterile filtered and 10% FBS was added to the media.

Human mesenchymal stromal/stem cells were resuspended into either growth medium with the addition of 10% FBS (GM) or the different conditioned media solutions that were prepared as described before with an appropriate cell density to obtain 500,000 hMSCs per IPN construct. The hydrogels were then cultured for either 1 (GM D1) or 21 days (GM D21, S-CM_0.5_, MA-CM_0.5_, and MA-CM_5_) in osteogenic medium (growth medium with the addition of 10% FBS and osteogenic factors) with a medium change twice a week. The complete experimental planning for this study is shown in Figure 1.

Either after 1 or 21 days, the osteogenic media was removed and the IPN was washed with PBS. The materials for biochemical assessment were cut in two and put into 1.5 mL microtubes for either papain or hydrochloric acid digestion. These microtubes were then stored overnight at −80 °C. The histology samples were fixed with 4% PFA for 4 h, dehydrated in 30% EtOH, and stored in 50% EtOH.

#### 2.7.1. DNA Analysis

A papain buffer extract (PBE) was made by adding 0.7098 g of dibasic anhydrous sodium phosphate (Sigma), 0.5998 g monobasic sodium phosphate (Sigma), and 1 mL of 500 mM EDTA to 90 mL of ultrapure water. The pH was brought to 6.5 by adding 38% HCl (Sigma). Ultrapure water was then added up to 100 mL and the solution was sterile filtered to remove dust particles. The papain digestion buffer was activated by adding 63 mg of L-cysteine hydrochloride hydrate (Sigma) together with 3.88 units/mL of papain enzyme to 40 mL of PBE. Samples were removed from the freezer and 472.5 μL of the activated papain buffer extract was added to each microtube before being put on a rocker for 18 h at 60 °C. Then, 27.5 μL of a 1 M sodium citrate solution was subsequently added to fully digest the IPN sample. The samples were then vortexed and centrifuged at 15,000 g for 10 min at 4 °C and the amount of DNA within these samples was obtained with a quant-it picogreen dsDNA assay kit (Invitrogen) by referring to set standards. The emission of both samples and standards was captured at 520 nm after excitation at 480 nm.

#### 2.7.2. Collagen Production

A hydroxyproline assay was performed to quantify the amount of collagen present in the samples as previously described [31]. Briefly, the papain digested sample was combined with PBE and 38% hydrochloric acid. The samples were then incubated at 110 °C for 18 h, cooled, centrifuged, allowed to dry in a fume hood and dissolved in ultrapure water. Hydroxyproline standards were made by serial dilution of the hydroxyproline stock solution. Both standards and samples were subsequently added to a 96-well plate in triplicate. A citrate stock buffer was made as the basis of the assay buffer and the chloramine T reagent which were added to both standards and samples in the 96-well plate. After the oxidation of hydroxyproline, (dimethylamino)benzaldehyde (DMBA) reagent was added and the absorbance of the plate was read at 570 nm. Qualification of the collagen content in the IPN samples was obtained through picrosirius red staining. The histology samples stored in 50% EtOH were further dehydrated in more concentrated EtOH and xylene baths before being embedded in paraffin wax. A microtome was used to slice the samples where after these were put in a water bath and mounted on a glass slide. After drying overnight, picrosirius red staining was applied and a bright-field microscope was used to image with a 2× and 10× objective.

#### 2.7.3. Mineral Production

A calcium assay kit (Sentinel Diagnostics, Milan, Italy) was used to quantify the amount of calcium present in the samples compared to standards which were made as serial dilutions of a 20 μg/mL and 100 μg/mL stock solution based on calcium chloride and hydrochloric acid solutions. Mineral depositions were also visualized through alizarin red staining of the sliced IPN samples. These were then subsequently imaged with a bright-field microscope using a 2× and 10× objective.

### 2.8. Statistical Analysis

The data obtained in this study was subjected to a statistical analysis with GraphPad Prism v8.2.0 (GraphPad Software, La Jolla, CA, USA) presented as mean ± standard deviation. The different experimental groups were compared using an ordinary one-way ANOVA with Tukey’s multiple comparison test. A *p*-value less than 0.05 was accepted as a significant difference between groups.

## 3. Results

### 3.1. Fabrication of an IPN Material with Mechanical Properties Mirroring the Bone Marrow/Osteoid Niche

Figure 2a gives an overview of the IPN hydrogel synthesis where the constant collagen concentration is mixed with different concentrations of alginate. Unconfined compressive testing was performed to quantify the compressive modulus of the developed polymer networks (Figure 2b). Addition of alginate results in a significant considerable increase in the modulus of the material. Whereas the modulus of collagen type I (COL-1) lies around 0.3 kPa, it is increased to around 25 kPa with the addition of 5 mg/mL of alginate (IPN 5). Further addition of alginate results in significantly increased moduli of around 60 kPa (IPN 10) and 85 kPa (IPN 15). To determine the influence of alginate addition to the microstructure of the materials, each IPN was observed under SEM (Figure 2c). The collagen type I network on its own possesses a fibrillar structure with small pore spaces between the fibers. Addition of 5 mg/mL of alginate reduces these pore spaces and tends to break up the fibrillar structure. When 10 mg/mL of alginate is added to the collagen type I network, the fibrillar microstructure is restored with enlarged pore spaces between the fibers making the structure more open. The pore spaces are further enlarged by the addition of 15 mg/mL of alginate. As the IPN 10 hydrogel possesses both a compressive modulus and a fibrillar porous architecture similar to that previously shown to enhance osteogenesis, this material was utilized for all further experimentation.

### 3.2. Paracrine Factors Released by Osteocytes Can Enhance Osteogenic Matrix Deposition by hMSCs in Monolayer Culture

In Figure 3a, an overview of the different media groups is given that will be used in subsequent experiments, i.e., growth media (GM), statically cultured osteocyte conditioned media (S-CM), and mechanically activated osteocyte conditioned media (MA-CM). The proliferation of hMSCs over 21 days was quantified through a picogreen dsDNA assay (Figure 3b). The addition of osteogenic supplements resulted in a significant 2.4-fold increase in proliferation of hMSCs when compared to standard growth medium (*p* < 0.05, n = 5). While an increase in hMSC proliferation was also seen following treatment with S-CM (1.6-fold) and MA-CM (1.7-fold) when compared to GM, this was not significant. This is likely due to the depletion of nutrients in the media during the conditioning procedure. There was also no difference in hMSC proliferation when comparing S-CM to MA-CM.

To determine the effect of osteocyte secreted factors on hMSC matrix deposition, we next analyzed collagen and matrix deposition after 21 days. As expected, the addition of osteogenic supplements resulted in a significant 4-fold (*p* < 0.05, n = 5) increase in collagen deposition (Figure 3c). An increase in collagen deposition following treatment with S-CM (2-fold) and MA-CM (2-fold) was also identified when compared to GM, but this was significantly less than that seen with GM^ost^. This trend mirrors that seen in the DNA analysis. Interestingly, mineral deposition was significantly altered by factors secreted by osteocytes (Figure 3d). Both the S-CM and MA-CM groups significantly enhanced mineral deposition over both GM and GM^ost^ (*p* < 0.05, n = 5). However, there was no difference in mineralization when comparing S-CM to MA-CM.

Taken together, this data demonstrates that paracrine factors released by osteocytes can enhance osteogenic matrix deposition by hMSCs in monolayer culture, and thus represent potential factors to enhance bone regeneration.

### 3.3. Osteocyte Derived Paracrine Factors Maintain hMSC Viability in Biomimetic IPN Hydrogels

We next looked to combine the paracrine factors released by osteocytes with the IPN materials to achieve a scaffold that recapitulates some of the biochemical and biophysical cues similar to that seen in the bone niche. Figure 4a schematically displays how both the hMSCs and the conditioned medium were incorporated into the IPN 10 material. A picogreen dsDNA assay was used to quantify the viability and proliferation of hMSCs within the IPN hydrogel over 21 days (Figure 4b). All groups show a significant decrease in DNA content compared to the day 1 group, demonstrating that these materials can have adverse effects on viability over long term culture. However, the addition of osteocyte secreted biochemical factors significantly improved cell viability approximately 2-fold (*p* < 0.05, n = 5) when compared to GM alone at day 21. As seen in 2D culture, no difference was seen between S-CM and MA-CM, nor was it influenced by CM concentration.

### 3.4. IPN Hydrogels Functionalised with Osteocyte Derived Biochemical Factors Enhance hMSC Osteogenic Matrix Deposition

We next examined whether the osteocyte derived biochemical factors could enhance osteogenic matrix deposition within biomimetic IPN 10 hydrogels. The picrosirius red collagen staining is evident across both the complete IPN hydrogel (Figure 5a, inset) and at a higher magnification (Figure 5a). Furthermore, at high magnification, punctate collagen staining is seen surround entrapped cells demonstrating collagen production by encapsulated cells. Interestingly there is no increase in collagen deposition over 21 days in the GM group but an increasing trend in deposition is seen following the addition of osteocyte secreted factors. The mechanically activated osteocyte conditioned media groups have an overall higher amount of staining across the hydrogels when analyzing the magnified images which is also confirmed in the quantification shown in Figure 5b. This increase in collagen deposition reaches significance in the higher concentration MA-CM compared to the growth media groups (*p* < 0.05, n = 5). The dose-dependent effect can also be observed when comparing both MA-CM groups. This data demonstrates the osteocyte derived factors can enhance hMSC collagen deposition in IPN hydrogels, although high concentrations are required.

Mineralization, as measured by Alizarin red staining, is evident across all groups by day 21 shown in Figure 5c. Interestingly, the GM IPN demonstrated poor mineralization within the center of the material while each of the osteocyte CM IPNs demonstrated good mineralization throughout, with high intensity staining observed surrounding encapsulated cells. Quantifying the mineral deposited, there is significant ~10-fold increase in calcium content after 21 days culture (Figure 5d). The osteocyte CM groups did not significantly influence the total concentration of mineral deposited when compared to GM at day 21 except for the MA-CM at 0.5 mg/mL which demonstrated a reduced total mineral. However, this was corrected when MA-CM concentration was increased to 5 mg/mL. Therefore, osteocyte derived factors do not increase overall quantity of mineral in IPN hydrogels but improves the distribution of mineral throughout the material.

## 4. Discussion

Due to the unmet clinical need for new materials to promote skeletal repair, this study aimed to produce novel biomimetic materials to enhance stem/stromal cell osteogenesis and bone repair by recapitulating aspects of the biophysical and biochemical cues found within the bone microenvironment. Utilizing a collagen type I–alginate interpenetrating polymer network we fabricated a material which mirrors the mechanical and structural properties of unmineralized bone, consisting of a porous fibrous matrix with a young’s modulus of 64 kPa, both of which have been shown to enhance MSC osteogenesis. Moreover, by combining this material with biochemical paracrine factors released by statically cultured and mechanically stimulated osteocytes, we furthered mirrored the biochemical environment of the bone niche, enhancing stromal/stem cell viability, differentiation and matrix deposition. Therefore, this biomimetic material represents a novel approach to promote skeletal repair.

Various cues have been shown to affect the lineage commitment of bone marrow stromal/stem cells. Collagen type I was selected for this study as this structural protein is the most abundant protein found in bone and exhibits a fibrillar architecture that facilitates cell adherence which has been shown to favor osteogenic differentiation [9,10]. However, collagen alone displays poor mechanical properties. By adding alginate to collagen type I, forming an interpenetrating polymer network, we were able to significantly improve the mechanical properties of the material in terms of compressive modulus. The IPNs containing 5 and 10 mg/mL alginate possess compressive moduli that lie within the range predicted to be optimal for osteogenesis [8]. The reported electrostatic complexation with alginate further adds to the roughness of the material on the microstructural level which has also been shown to be favorable and hence further adds to the stimulating environment for osteogenic differentiation [14]. Increasing the alginate content of the IPN also resulted in a more open network structure. This can be explained by a reduced number of available nucleation sites for collagen fibrillogenesis due to the choice for a low concentration of collagen type I present in the IPN material as well as the formation of an electrostatic complex with alginate leading to the inhibition of the branching lateral addition of collagen molecules [32]. It was hypothesized that this more open structure would facilitate cell migration and the diffusion of nutrients. Ultimately, the IPN with 10 mg/mL alginate was chosen since it has the highest ability to incorporate both biophysical and biochemical cues present in the bone marrow stromal cell niche.

In vivo, mechanical loading leads to bone formation by recruiting or activating osteoprogenitor cells from the marrow stroma and along the bone surface which subsequently differentiate into bone forming osteoblasts [33,34]. Osteocytes have been shown to play a key role in this bone mechanoadaptation [26,35]. To decipher the potential regenerative properties of biochemical factors released by osteocytes, we collected the secretome of statically cultured and mechanically stimulated osteocytes and utilized this to treat human MSCs and evaluate cell viability/proliferation and osteogenic matrix deposition in 2D culture. Conditioned media, either static or mechanically activated, was found to have a minimal effect on stem cell proliferation, which is consistent with previous findings by Birmingham et al. [23]. This is partially in contrast to Brady et al. who identified a significant increase in murine MSC proliferation following treatment in mechanically activated osteocyte conditioned medium. [25]. However, this discrepancy in proliferative effect may be due to the different species, the different mechanical stimulus that was applied (a rocking platform versus an orbital shaker), the difference in duration of mechanical activation (24 h instead of the 2 h time frame used in this experiment), and finally due to the use of a shorter time point at which proliferation was quantified (day 3 versus day 21). This minimal effect of the osteocyte secretome was also seen in terms of collagen deposition. However, this is in contrast to the mineral production of the conditioned media groups which was significantly increased compared to controls. This is consistent with data obtained by Heino et al. for static osteocyte conditioned medium [24]. Brady et al. found that the normalized calcium concentration of both the mechanically activated osteocyte conditioned medium and the positive control was significantly upregulated compared to static osteocyte conditioned medium and the negative control group [25]. This discrepancy arises possibly because no osteogenic factors were added in their conditioned media groups or, as discussed earlier, due to either a different mechanical activation or the use of another cell type. In conclusion, the main effect of adding osteocyte conditioned medium in terms of matrix deposition can be found in the production of mineral deposits. However, the orbital shaker mechanically activated osteocyte conditioned medium shows the same mineral production as the one observed with static osteocyte conditioned medium, suggesting that a more robust mechanical stimulation regimen is required to harness the benefits of physical loading previously identified.

To elucidate the combined effect of the cues presented in the paracrine signaling of osteocytes and the cues incorporated into the biomaterial, an osteogenic differentiation study was conducted that evaluated both cell viability as well as matrix deposition in 3D cell culture. It was hypothesized that the experimental biomaterial groups that incorporated the conditioned medium would have a more favorable effect on the osteogenic differentiation. Moreover, it was speculated that the concentrated mechanically activated conditioned media group would lead to an enhanced osteogenic differentiation. The number of cells present after 21 days quantified through the amount of DNA content is significantly decreased compared to the negative control at day 1. This could be attributed to a lack of nutrients especially in the center of the hydrogel which can be seen in the histology images of both the collagen production and more pronounced in the mineral production for the growth medium group quantified at day 21. The conditioned media groups still show a significant decrease compared to day 1 although significantly less pronounced than the one identified with the negative control at day 21. This shows that the addition of conditioned medium to the hydrogel positively influences cell survival but is not able to completely turn the issue of cell death in the core of the hydrogel around.

Combining the biophysical cues within collagen based IPN hydrogels with biochemical cues derived from osteocytes, we developed a material that recapitulates aspects of the bone microenvironment, enhanced cell viability and osteogenic matrix deposition. While the IPN material presents a mimetic architecture, it was evident that cell death occurred particularly within the center of this construct over a 21-day period which is indicative of poor nutrient transport. Interestingly, the addition of biochemical factors released by osteocytes significantly improved cell viability. This positive influence of osteocyte derived factors is further evident when examining matrix deposition. Collagen deposition is significantly enhanced by concentrated mechanically activated osteocyte derived factors. Moreover, a uniform distribution of mineralization was clear throughout the material while mineralization is absent in the core of materials lacking osteocyte derived factors. While the increase in mineralization relative to GM controls seen in 2D culture was not found in the IPN materials, this is likely due the inherent osteogenic capacity of the collagen based IPN hydrogel. Taken together this data demonstrates that mimicking aspects of the bone microenvironment can be utilized as a strategy to enhance MSC osteogenesis and bone repair.

There are a number of limitations in this study. For example, the orbital shaker used here was not able to optimally activate osteocytes since the effect of the mechanically activated conditioned medium, despite some minor non-significant trends, did not differ significantly from its static counterpart, as has been previously shown using alternative bioreactors. Although shear stress values reported with the orbital shaker vary between 0.02 and 1.3 Pa, the delivered shear stress distribution might be too low and irregular to achieve a sufficient mechanical activation [36]. Future work should look at a new type of bioreactor that is able to have a higher control over the delivered shear stress distribution, whilst also enabling the generation of large volumes of media. Concentrating the conditioned media should also be explored further as a mechanism to stimulate osteogenesis next to an optimal mechanical activation.

## 5. Conclusions

In this study we recapitulated aspects of the biophysical and biochemical cues found within the bone microenvironment in a material as a strategy to address the clinical need for novel materials to enhance bone repair. Utilizing a collagen based interpenetrating polymer network we fabricated a material which mirrors the mechanical and structural properties of unmineralized bone, consisting of a porous fibrous matrix with a young’s modulus of 64 kPa, both of which have been shown to enhance MSC osteogenesis. Moreover, by combining this material with biochemical paracrine factors released by osteocytes, we furthered mirrored the biochemical environment of the bone niche, enhancing stromal/stem cell viability, differentiation and matrix deposition. Therefore, this biomimetic material represents a novel approach to promote skeletal repair.

## Figures and Tables

**Figure 1 materials-13-01690-f001:**
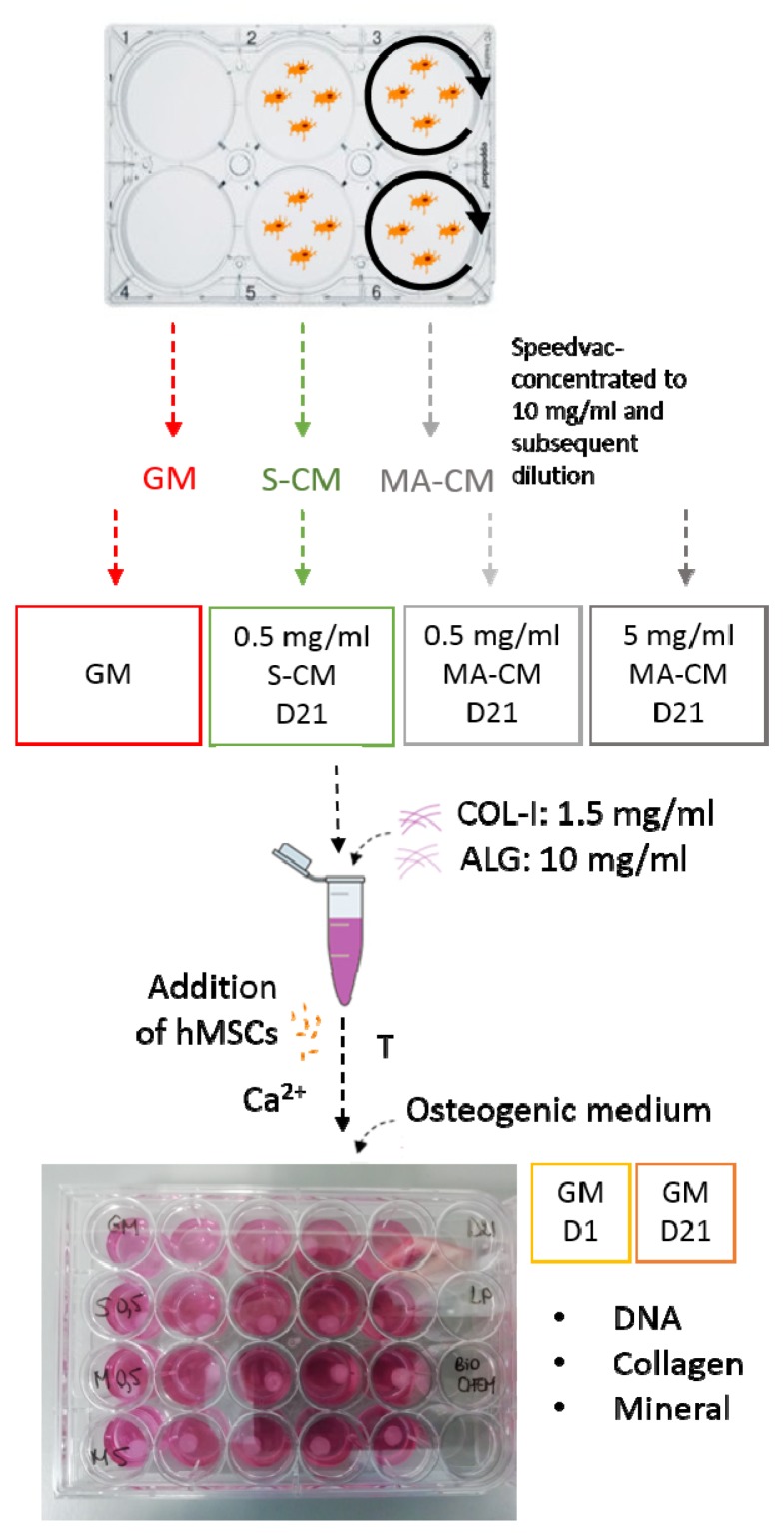
Experimental outline investigating the effect of osteocyte secreted factors on human mesenchymal stromal/stem cells (hMSC) osteogenesis within interpenetrating polymer network (IPN) hydrogels comprising collagen type I (COL-I) and alginate (ALG). Five different groups are investigated: GM = growth medium (α-modification of Eagle’s minimal essential medium (α-MEM) supplemented with 10% fetal bovine serum (FBS), 1% Pen/Strep, and 1% L-glutamine), 0.5 mg/mL statically cultured osteocyte conditioned media (S-CM) supplemented with 10% FBS, 0.5 mg/mL mechanically activated osteocyte conditioned media (MA-CM), and 5 mg/mL MA-CM supplemented with 10% FBS. hMSCs were resuspended into these media groups and incorporated into the COL-I/ALG hydrogel subsequently exposed to thermal gelation (T), ionic crosslinking (Ca^2+^), and cultured for 21 days in osteogenic medium. DNA content, collagen, and mineral deposition was quantified at D1 and D21, n = 5.

**Figure 2 materials-13-01690-f002:**
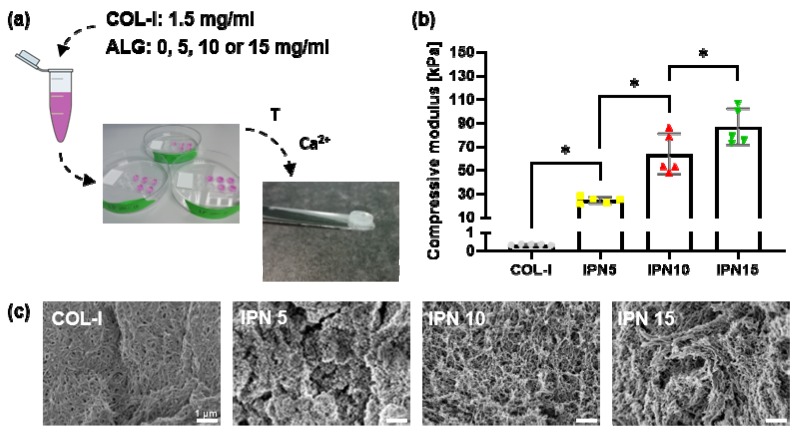
Fabrication of an IPN material mirroring the bone marrow/osteoid niche (**a**) Overview of the IPN synthesis with a 1.5 mg/mL collagen type I (COL-I) concentration and a varying alginate (ALG) concentration of 0, 5, 10, and 15 mg/mL subjected to a collagen type I gelation phase at 37 °C (T) for 30 min next to an ionic crosslinking (Ca^2+^) phase of alginate at 37 °C for 3 h; (**b**) unconfined compressive modulus compared for four different materials: a collagen type I hydrogel (COL-I), a COL-I/ALG IPN hydrogel with 5 mg/mL alginate (IPN 5), a COL-I/ALG IPN hydrogel with 10 mg/mL alginate (IPN 10), and a COL-I/ALG IPN hydrogel with 15 mg/mL alginate (IPN 15) (n = 5); (**c**) SEM images of COL-I, IPN 5, IPN 10, and IPN 15 (scale bar: 1 μm).

**Figure 3 materials-13-01690-f003:**
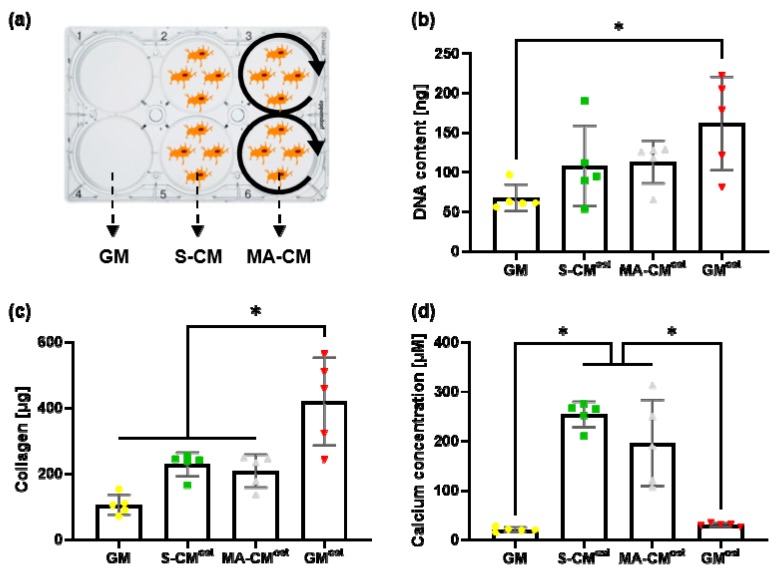
Effect of paracrine factors released by osteocytes on hMSC behavior in 2D cell culture. (**a**) Overview of the different media groups including growth medium (GM), conditioned medium collected from statically cultured osteocytes (S-CM) and conditioned medium collected from mechanically activated osteocytes (MA-CM); (**b**) quantification of the DNA content through a picogreen dsDNA kit for hMSCs treated with GM, S-CM^ost^, MA-CM^ost^, and the growth medium group supplemented with osteogenic factors (GM^ost^) (n = 5); (**c**) quantification of collagen deposition through the amount of bound picrosirius red staining for each medium group (n = 5); (**d**) quantification of mineral deposition through the amount of bound alizarin red staining for every medium group (n = 5).

**Figure 4 materials-13-01690-f004:**
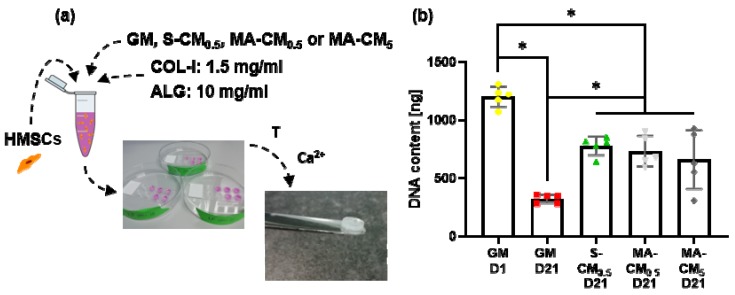
Effect of osteocyte derived paracrine factors on cell viability in IPN 10 hydrogels. (**a**) Schematic overview of the different media groups supplemented with 10% FBS and osteogenic factors including growth medium quantified at both day 1 and day 21 after incorporating the hMSCs into the IPN hydrogel (GM D1 and GM D21), Speedvac-concentrated static osteocyte conditioned medium diluted to a protein content of 0.5 mg/mL quantified at day 21 after hMSC incorporation (S-CM_0.5_ D21), Speedvac-concentrated mechanically activated osteocyte conditioned medium diluted to a protein content of 0.5 mg/mL quantified at day 21 after hMSC incorporation (MA-CM_0.5_ D21), and Speedvac-concentrated mechanically activated osteocyte conditioned medium diluted to a protein content of 5 mg/mL quantified at day 21 after hMSC incorporation (MA-CM_5_ D21); (**b**) quantification of the DNA content through a picogreen dsDNA kit for the different media groups (n = 5).

**Figure 5 materials-13-01690-f005:**
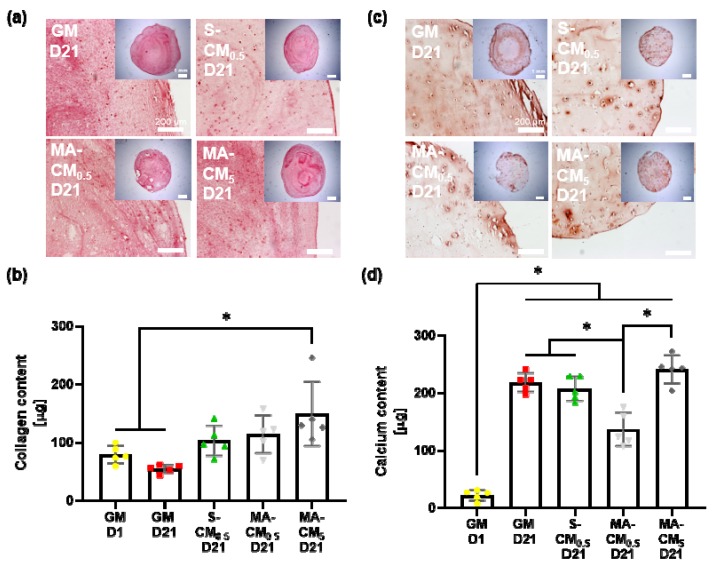
IPN hydrogels functionalized with osteocyte derived biochemical factors enhance hMSC osteogenic matrix deposition. (**a**) Collagen production qualified through picrosirius red staining for the different media groups supplemented with 10% FBS and osteogenic factors including growth media GM D1 and GM D21, osteocyte conditioned medium at 0.5 mg/mL quantified at day 21 after hMSC incorporation (S-CM_0.5_ D21), mechanically activated osteocyte conditioned medium at 0.5 mg/mL and 5 mg/mL quantified at day 21 after hMSC incorporation (MA-CM_0.5_ D21 and MA-CM_5_ D21) (scale bar: 200 μm, scale bar inset: 1 mm); (**b**) collagen content quantified with a hydroxyproline assay for every medium group (n = 5); (**c**) mineral production qualified through alizarin red staining for the different media groups (scale bar: 200 μm, scale bar inset: 1 mm); (**d**) mineral content quantified with a calcium assay for every medium group (n = 5).

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
