# Peer review of "Utilizing Osteocyte Derived Factors to Enhance Cell Viability and Osteogenic Matrix Deposition within IPN Hydrogels"

_materials, 2020, doi:10.3390/ma13071690_

Round 1

Reviewer 1 Report

Dear Authors,

I have gone though your manuscript and impressed with your work. I have some suggestions which will enhance the readability of the manuscript.

1- Title of the manuscript too long and has no connection with research have been performed in this study.

2- Introduction need to correlate with objective of the manuscript.

3- Some new recent reference need to be incorporated and as author want to publish their wor in very prestigious journal Materials,i will strongly recommend to add some Materials journal reference need to be included.

4- Include some recent references in Materials method

5- The cell viability data should be included.

Author Response

Dear reviewer,

Thank you for reviewing our paper entitled “Utilizing osteocyte derived factors to enhance cell viability and osteogenic matrix deposition within IPN hydrogels”. We have responded to all advised revisions and have included the list of specific changes, listed as follows (page #, paragraph #), according to your comments below. All the edits (indicated in red) of the different reviewers can also be found in the manuscript in attachment.  

Regards,

Laurens Parmentier, Mathieu Riffault and David Hoey

Dear Authors,

I have gone through your manuscript and impressed with your work.

We thank the reviewer for their kind words.

I have some suggestions which will enhance the readability of the manuscript:

1- Title of the manuscript too long and has no connection with research have been performed in this study.

We thank the reviewer for this recommendation, the title has been edited accordingly.

(page 1, paragraph 1) replaced: ‘Combining bone mimetic biophysical and biochemical cues to enhance cell viability and osteogenic matrix deposition within IPN hydrogels’ with ‘Utilizing osteocyte derived factors to enhance cell viability and osteogenic matrix deposition within IPN hydrogels’

2- Introduction need to correlate with objective of the manuscript.

We thank Reviewer 1 for their recommendation. The objective of this work was to replicate native biophysical and biochemical cues found within bone to better engineer a synthetic material to promote bone repair. Therefore, we have structured the introduction to introduce the limitations of bone grafts, the importance of biophysical and biochemical cues delivered by materials which mimic the native bone, the biophysical and biochemical cues delivered by resident cells within native bone, and finish by stating our objective to combine all these cues to engineer an alternative approach to bone repair. We therefore feel that our introduction does correlate with the objectives of our study. Having said this we have significantly rewritten the introduction to better articulate our goal (highlighted in red).

3- Some new recent references need to be incorporated and as author want to publish their work in very prestigious journal Materials, I will strongly recommend you add some Materials journal reference need to be included.

We thank the reviewer for raising an interesting point. Therefore, we have included a recent Materials journal reference which elaborates on the need for a 3D cell culture environment which incorporates several biophysical cues as is the case in the native extracellular matrix.

(page 2, paragraph 2) replaced: ‘Therefore, a synthetic alternative to autograft should aspire to recapitulate the structural and physical properties of bone ECM.’ with ‘Therefore, there is a need for a synthetic alternative to autograft which aspires to recapitulate the various structural and physical properties of bone ECM in a 3D environment [1].’

4- Include some recent references in Materials method

We thank the reviewer for this recommendation and have included recent references in the materials and methods section regarding cell culture and collagen and mineral production quantification.  

(page 3, paragraph 4) replaced: ‘The MLO-Y4 cell line (Kerafast) is a murine derived osteocyte cell line that was cultured on 0.15 mg/ml rat tail collagen type I (Sigma)-coated culture flasks with α-modification of Eagle’s minimal essential medium (α-MEM) (Labtech) supplemented with 5% fetal bovine serum (FBS) (Labtech), 5% calf serum (CS) (Sigma), 1% penicillin/streptomycin (Pen/Strep) (Sigma) and 1% L-glutamine (Sigma).’ with ‘The MLO-Y4 cell line (Kerafast) [2] is a murine derived osteocyte cell line that was cultured, as previously described [3], on 0.15 mg/ml rat tail collagen type I (Sigma)-coated culture flasks with α-modification of Eagle’s minimal essential medium (α-MEM) (Labtech) supplemented with 5% fetal bovine serum (FBS) (Labtech), 5% calf serum (CS) (Sigma), 1% penicillin/streptomycin (Pen/Strep) (Sigma) and 1% L-glutamine (Sigma).

(page 4, paragraph 4) added: ‘Collagen was stained and quantified through a protocol previously described [4,5].’  

(page 4, paragraph 5) added: ‘Mineral deposits were stained and quantified through a protocol previously described [4,5].’

5- The cell viability data should be included.

We have assessed cell viability in terms of DNA content at Days 1 and 21 and this data is presented in Figures 3 for the 2D analysis and Figure 4 for the 3D analysis.

References

  1. Werner, M.; Kurniawan, N.A.; Bouten, C.V. Cellular geometry sensing at different length scales and its implications for scaffold design. Materials 2020, 13, 963.
  2. Kato, Y.; Windle, J.J.; Koop, B.A.; Mundy, G.R.; Bonewald, L.F. Establishment of an osteocyte‐like cell line, MLO‐Y4. Journal of bone and mineral research 1997, 12, 2014-2023.
  3. Rosser, J.; Bonewald, L.F. Studying osteocyte function using the cell lines MLO-Y4 and MLO-A5. In Bone research protocols, Springer: 2012; pp. 67-81.
  4. Stavenschi, E.; Corrigan, M.A.; Johnson, G.P.; Riffault, M.; Hoey, D.A. Physiological cyclic hydrostatic pressure induces osteogenic lineage commitment of human bone marrow stem cells: a systematic study. Stem Cell Research & Therapy 2018, 9, 276, doi:10.1186/s13287-018-1025-8.
  5. Corrigan, M.A.; Coyle, S.; Eichholz, K.F.; Riffault, M.; Lenehan, B.; Hoey, D.A. Aged Osteoporotic Bone Marrow Stromal Cells Demonstrate Defective Recruitment, Mechanosensitivity, and Matrix Deposition. Cells Tissues Organs 2019, 207, 83-96.

Reviewer 2 Report

This manuscript reports the development of an interpenetrating network (IPN) hydrogel based on collagen type I combined with alginate. Authors describe changes of the microstructure and compressive modulus of the materials with the addition of increasing concentrations of alginate, as expected from the literature. This allowed authors to obtain a material replicating the architecture and mechanical properties of unmineralized bone. Further combining the biomaterial with osteocyte-derived conditioned media enhanced the osteogenic response of encapsulated mesenchymal stem cells. The introduction and experimental section are clearly described. Keywords could be more specific/better defined, as mechanobiology, for instance, does not reflect the scope of the study. A schematics of the experimental setup would be helpful to understand cellular experiments. In addition, results regarding pore size of the biomaterials and cell viability (live/dead assay for 3D culture) could be added to further support the discussion regarding cell death. The discussion clearly addresses the main outcomes and limitations of the study.

Author Response

Dear reviewer,

Thank you for reviewing our paper entitled “Utilizing osteocyte derived factors to enhance cell viability and osteogenic matrix deposition within IPN hydrogels”. We have responded to all advised revisions and have included the list of specific changes, listed as follows (page #, paragraph #), according to your comments below. All the edits (indicated in red) of the different reviewers can also be found in the manuscript in attachment.

Regards,

Laurens Parmentier, Mathieu Riffault and David Hoey

This manuscript reports the development of an interpenetrating network (IPN) hydrogel based on collagen type I combined with alginate. Authors describe changes of the microstructure and compressive modulus of the materials with the addition of increasing concentrations of alginate, as expected from the literature. This allowed authors to obtain a material replicating the architecture and mechanical properties of unmineralized bone. Further combining the biomaterial with osteocyte-derived conditioned media enhanced the osteogenic response of encapsulated mesenchymal stem cells. The introduction and experimental section are clearly described. The discussion clearly addresses the main outcomes and limitations of the study.

We thank the reviewer for their kind overview.

Keywords could be more specific/better defined, as mechanobiology, for instance, does not reflect the scope of the study.

We thank the reviewer for this recommendation, the keywords have been edited accordingly.

(page 1, paragraph 1) replaced: ‘bone; mechanobiology; biomaterials; osteocyte; IPN’ with ‘bone; collagen type I, alginate, conditioned medium, viability, MSC, osteogenesis’

A schematics of the experimental setup would be helpful to understand cellular experiments.

We thank the reviewer for this recommendation, an extra figure explaining the experimental planning of the ‘IPN + conditioned medium’ osteogenesis study has been added (new Figure 1, with caption below). All other figure numbers have been updated.

Figure 1. Experimental outline investigating the effect of osteocyte secreted factors on hMSC osteogenesis within IPN hydrogels. Five different groups are investigated: GM = growth medium (α-MEM supplemented with 10% FBS, 1 % Pen/Strep and 1 % L-glutamine), 0.5 mg/ml S-CM supplemented with 10% FBS, 0.5 mg/ml MA-CM and 5 mg/ml MA-CM supplemented with 10% FBS. hMSCs were resuspended into these media groups and incorporated into the COL-I/ALG hydrogel subsequently exposed to thermal gelation (T), ionic crosslinking (Ca2+) and cultured for 21 days in osteogenic medium. DNA content, collagen and mineral deposition was quantified at D1 and D21, n=5.

In addition, results regarding pore size of the biomaterials and cell viability (live/dead assay for 3D culture) could be added to further support the discussion regarding cell death.

We kept the discussion of the microstructure of the interpenetrating polymer networks qualitative since 2D pore size measurements on a 2D image of a 3D surface might give inaccurate results. However, the SEM images give an indication of pore sizes throughout the material.  

We have assessed cell viability in terms of DNA content at Days 1 and 21 and this data is presented in Figures 3 for the 2D analysis and Figure 4 for the 3D analysis. Live/Dead studies were not performed in this study and unfortunately due to the COVID-19 shutdown we are not currently in a position to redo these experiments.

Reviewer 3 Report

Dear Editor,

the paper by Parmentier and Hoey entitled “Combining bone mimetic biophysical and biochemical cues to enhance cell viability and osteogenic matrix deposition within IPN hydrogels” comprehensively elucidates a novel approach to promote skeletal repair. The subject fits the journal scope and, in my opinion, is interesting for the scientific community. The MS is well written and all the explanations are given.

From the scientific point of view, the work is OK.

The MS deserves publication as it is.

Author Response

Dear reviewer,

Thank you for reviewing our paper entitled “Utilizing osteocyte derived factors to enhance cell viability and osteogenic matrix deposition within IPN hydrogels”. We would also like to thank you for your kind overview. We have responded to all advised revisions and all the edits (indicated in red) of the other reviewers can also be found in the manuscript in attachment.

Regards,

Laurens Parmentier, Mathieu Riffault and David Hoey

Reviewer 4 Report

In this comprehensive study, the authors designed and tested a novel biomaterial that might be useful as a bone substitute. As bone regeneration remains challenging and autografts have well known disadvantages novel biomaterials that can be used in clinical routine are needed. Thus, I have read the manuscript with great interest. 

Overall the study was very well designed, the individuell steps are well-thought off and thus the results are robust. The findings are based on the results and are novel and the conclusions are supported by the findings and results. All in all this is an excellent study and I have no concerns. I especially enjoyed the presentation of the graphs, as the schematics help to interpret and understand the individual datasets much faster and better. 

Author Response

(The authors gave the same response as above.)
